# Sector-Wide Approach (SWAp) in Healthcare—A Mixed-Methods Assessment of Health SWAps in Nepal and Bangladesh

**DOI:** 10.3390/ijerph21121682

**Published:** 2024-12-17

**Authors:** Steffen Flessa

**Affiliations:** Department of Business Administration and Health Care Management, University of Greifswald, 17489 Greifswald, Germany; steffen.flessa@uni-greifswald.de

**Keywords:** aid effectiveness, Bangladesh, healthcare financing, Nepal, sector-wide approach, SWAp

## Abstract

Background: The sector-wide approach (SWAp) is an instrument of cooperation between external development partners (EDPs) and the government of a country. Its main purpose is the coordination, alignment and harmonisation of activities between EDPs and between EDPs and the respective government by covering the entire sector with one major programme. Methods: The main objective of this paper is to analyse the performance of the SWAps in two countries and draw conclusions about the appropriateness of SWAps as financing instruments in the healthcare sector under certain conditions. This paper analyses the development and achievements of the SWAp in healthcare of Bangladesh and Nepal in order to gain insights into the development and relevance of SWAps in the healthcare sectors of low- and lower-middle-income countries in general. We scrutinised the respective documents and conducted qualitative interviews with key stakeholders of the country’s sectors. The design of the questionnaires and the analysis of the interviews were built utilising a framework model reflecting the DAC criteria of development cooperation and the principles of the Declarations of Paris and Accra. Findings: The SWAps in Nepal and Bangladesh began rather early and cover about 20 years of cooperation. The components and interventions of SWAps were quite relevant for the health of the population, and their implementation was effective and efficient. The cohesion between partners strongly improved. However, for both countries, the interview partners do not perceive SWAps as the future of healthcare financing. Conclusions: SWAps were an appropriate instrument of cooperation between the respective governments and EDPs for almost two decades. However, as the share of government budgets in the sector finance has strongly increased and the management capacity of the respective ministries has gone up, there will come a point in time where EDPs can focus more on financing and implementing innovations instead of standard care.

## 1. Introduction

Global development assistance for health (DAH) amounts to some 70 billion US$ (2021) p.a. and has increased strongly in the last several years (11% p.a. 2011–2019) [1]. The strongest expansion was seen during the COVID-19 pandemic when DAH grew by 43.9% between 2019 and 2020, but even the absolute DAH in 2019 of 43 billion US$ was impressive. On a global level, DAH is almost negligible (1991: 0.3% of total healthcare expenditure; 2019: 0.5%), but for some countries external aid contributes more than 50% of the total healthcare expenditure. Micronesia (73%), South Sudan (64%), Zimbabwe (56%) and Mozambique (52%) finance their healthcare systems predominantly by foreign assistance. In Sub-Saharan Africa, 13.1% of total healthcare expenditure is financed by external health assistance. This rate is higher than in any other region. The two focus countries of this paper receive 10.5% (Nepal) and 5.4% (Bangladesh) external help of their total current health expenditure [2]. Consequently, the effectiveness and efficiency of DAH are crucial for the quality of life and survival of millions of people.

Table 1 exhibits different concepts of DAH. Traditionally, most external development partners (EDP) initiated projects of limited scope and time, e.g., establishing a hospital. Most of these projects were hardly coordinated with other projects and with healthcare activities of the respective government. It was soon realised that many healthcare problems cannot be addressed by stand-alone projects or a discontinuous series of projects. Instead, development programmes are required which run until a problem is solved or until the effort is taken over by the country’s healthcare services. Prevention (e.g., the Aids Control Programme), training or running certain curative facilities can still be initiated and operated by EDPs, but they require more long-term coordination, in particular with the government of the respective country.

The sector-wide approach (SWAp) coordinates all healthcare activities of the government and EDPs as “… an approach to a locally-owned program for a coherent sector in a comprehensive and coordinated manner, moving towards the use of country systems. SWAps represent a … shift in the focus, relationship and behaviour of donors and Governments. They involve high levels of donor and country coordination for the achievement of programme goals, and can be financed through parallel financing, pooled financing, general budget support, or a combination” [3].

Ideally, all partners (i.e., government and all EDPs) contribute to a single basket from which all healthcare activities are paid for, without earmarking specific donations for specific activities. Sometimes this is not possible, so that the basket-fund is supplemented by a single-donor trust fund that still contributes to the sector budget, but with restrictions on how it can be used.

**Table 1 ijerph-21-01682-t001:** Conceptions of development assistance for health.

Method	Horizon	Content	Ownership	Target	Coordination	Financing	Risk
Project	fixed	tailor-made intervention with limited scope	can be external development partner	specific challenge	limited	one source or consortium	poor coordination and alignment, poor sustainability, poor ownership
Programme	until problem is solved or taken over	tailor-made intervention with broader scope	can be external development partner	specific challenge	limited	one source or consortium	poor coordination and alignment, medium sustainability, poor ownership
Sector-wide programme	phases of a continuous sector programme	entire sector	government	entire health sector	completely within health sector	basket financing or single-donor trust fund	limited control by external development partners
General budget support	annual support	entire government	government	all sectors	completely with all sectors	non-earmarked budget	very limited control by external development partners

Source: author, based on [4,5].

While the SWAp is limited to a certain sector (usually health or education), the general budget support provides funds for a government without limitation to a certain sector. The control by the EDP is very low, i.e., funds might be used in a manner which does not reflect the value system of the EDP.

For some time, SWAps were seen as the magic bullet of DAH [6]. In particular, after the formulation of the Paris (2005) and Accra (2008) Declarations calling for more aid effectiveness by aligning development assistance and the ownership of programmes by the partner country [7], SWAps were fostered and implemented in many countries in Sub-Saharan Africa and Asia. Several evaluations have shown that the alignment between EDPs and between EDPs and the country’s government has improved, while the feeling of “our programme” in the respective country increased through SWAps [8,9]. Nevertheless, SWAps are on the decline, i.e., more EDPs have returned to specific programmes with a lower degree of coordination and alignment. This calls for an analysis of the underlying reasons and for an assessment of the strengths and weaknesses of health SWAps.

This paper has the objective of providing insights into the development and relevance of SWAps in the healthcare sector of low- and lower-middle-income countries, using Nepal and Bangladesh as examples. Based on these examples, we would like to determine the pros and cons of health SWAps and analyse the conditions of successful SWAps, i.e., the time where SWAps are most effective, efficient and superior to other forms of development assistance for health (“best time of SWAps”). In the following methods section, we will introduce the SWAps and the country setting as well as the methodology of the study. Afterwards, relevant statistics and the results of qualitative interviews will be described. In the discussion, these results will be used for an assessment based on the perception of the interviewees.

## 2. Methods

### 2.1. Setting

This paper is based on surveys in Nepal and Bangladesh. We combine the analyses of both countries in one paper in order to learn from similarities and differences so that the conclusions can be based on a higher degree of evidence.

Both countries have become lower-middle-income countries within the last decade. The Democratic Republic of Nepal has a GDP of 1337 US$ p.c. (current 2022) with a high growth rate (2022: 5.6%) [2]. The Human Development Index (HDI) improved from 2014 to 2021 from 0.56 to 0.60 [10]. A major challenge in the country is the tremendous disparity between provinces, urban and rural regions, social and ethnic groups, castes and gender concerning income, education, quality of life and health [11,12,13].

Health is a focus area of the government of Nepal. This has prompted the development of several national, sector-wide programmes. The first SWAp was the National Health Sector Programme I (NHSP-I, 2005–2010), with a budget of 620 million US$ [14], followed by NHSP-II (2010–2016), with a budget of 1.2 billion US$, and the National Health Sector Strategy (NHSS 2016–2022), with a total of 2.662 billion US$ [15]. These national programmes were a response of the government of Nepal as well as its external development partners to the comparably poor state of health of the population. Between 2010/11 and 2019/20, the share of health expenditure covered by EDPs strongly declined from 41 to 21%. During the COVID-19 pandemic, the respective expenditure increased again to 63% (2020/21) and 55% (2021/22) [16]; however, this appears to have been a special COVID-19 effect.

The situation in Bangladesh is similar, although the country is slightly richer with a GPD of 2688 US$ (current p.c.) and a growth rate of 7.1% [2] resulting in a higher HDP of 0.66 (2021) [10]. Bangladesh is the country with the highest population density worldwide (1313 inhabitants per sq. km) if we disregard city-states like Singapore. Urbanisation and the enormous growth of mega-cities such as Dhaka are a major challenge for the country.

Bangladesh has a rather homogenous population with respect to ethnicity and religion, but disparities between poor and rich are wide and pose a challenge. The poorer 50% of the population owns only 4.8% of all wealth and receives merely 17.1% of all income [17]. In peripheral regions, women have a lower average income [18], while women and children have a lower education [19,20], nutrition [21] and health status [22].

The first sector-wide programme in Bangladesh was the “Health and Population Programme” (HPSP, 1998 to 2003), which had a total budget of 2.2 billion US$. This was followed by the “Health, Nutrition and Population Sector Programme” (HNPDP) which was implemented from 2003 to 2011 with a budget of 5.4 billion US$, and subsequently by the “Health, Population and Nutrition Sector Development Programme” (HPNSDP), which ran from 2011 to 2017 with a total budget of 7.7 billion US$. This programme was succeeded by the “Health, Population and Nutrition Sector Programme” (HPNSP) from 2017 to 2022, with a total budget of 14.7 billion US$.

Figure 1 shows the timelines of the SWAps, including the budget per inhabitant of the respective country. We took the average population of each respective programme as a denominator [2]. It can be seen that Bangladesh started earlier with better-financed SWAps, i.e., which have higher budgets per capita. This difference can be explained by the higher GNI p.c. Otherwise, the SWAps are rather similar.

### 2.2. Mixed-Methods Evaluation

In this study, we used a mixed-methods approach to evaluate the SWAps implemented in Nepal and Bangladesh. We performed a secondary data analysis of documents and statistics of EDPs and the respective governments, in particular programmes, demographic and health surveys, health management information systems (HMIS), policy reports and healthcare financing strategies, as well as mid-term and final evaluations of the programmes. Based on these reports, we could assess the objective outcomes of the programmes. However, it is impossible to evaluate whether these outcomes were a consequence of the implementation of the SWAps. Consequently, we also had to collect qualitative evidence through key informant interviews and with leaders of EDPs, ministries and healthcare facilities in both countries as well as focus group discussions (FGD) with selected leaders of EDP and ministries. We conducted these interviews with the key decision-makers of the respective healthcare sectors in order to retrieve their perceptions, assessments and inclinations towards SWAps. As we wanted to assess these SWAps in Nepal and Bangladesh in order to gain insights into the appropriateness and “best time” for SWAps in low- and middle-income countries, we had to find these perceptions. These “soft facts” determine the decisions on development assistance for health and the most appropriate healthcare financing instrument at least as much as the “hard facts” of performance indicators and outcome statistics.

For Nepal, we conducted 10 interviews with EDPs, 9 interviews with co-workers of the Ministry of Health (MoH) and Ministry of Finance (MoF), 24 interviews with leaders of hospitals and health centres, 6 interviews with warehouse managers and 6 with consultants (e.g., procurement of contraceptives). For Bangladesh, we interviewed 10 EDPs, 15 members of the MoH and MoF, 25 managers of healthcare facilities, two academic scholars (professors of healthcare management in Bangladesh) and one consultant (procurement of contraceptives). The interviews were personally conducted by the author in April/May 2023 (Nepal) and January/February 2024 (Bangladesh). Each interview lasted, on average, 60 min. Interviews with EDPs were partly done as video conferences, and all other interviews were conducted face-to-face in the respective workplaces of the interviewees. The results were presented to selected interviewees based on the DAC criteria (see below) in order to obtain their opinion and analyse their interaction in the light of the performance of the respective SWAp in Nepal and Bangladesh. These workshops (one per country) were organised as FGDs under the structured leadership of one EDP.

In accordance with Mayring [23], we developed categories based on an evaluation framework exhibited in Figure 2. The theoretical basis of this model is the work of [24]. In the core, we see the five dimensions of the Declaration of Paris of effective aid, namely (1) ownership of partner countries, (2) alignment of EDPs, (3) harmonisation among EDPs, (4) managing for results and (5) mutual accountability. These universal principles are influenced by a frame of parameters for successful implementation for the case of a health SWAp, i.e., (a) government commitment to the SWAp, (b) legitimacy, (c) accountability and (d) leadership of the government, (e) focus on systems strengthening, (f) institutional development and (g) EDP commitment to the SWAp. The evaluation of these parameters is based on the DAC criteria of development cooperation, i.e., relevance, cohesion, effectiveness, efficiency, impact and sustainability [25].

The DAC criteria build on each other. The interventions with a certain input (e.g., budget) are supposed to produce certain outputs that will generate outcomes. It is assumed that these outcomes will have an impact on the general objectives of the society or the healthcare system. For instance, procurement of contraceptives (input) will increase the availability of contraceptives in health centres (output), which is the prerequisite for contraceptives being used (outcome). This might reduce the total fertility rate of the population (impact). The input is a necessary but insufficient condition of the output, which is a necessary but insufficient condition of the outcome, which is a necessary but insufficient condition for the impact. The criteria of efficiency compare the input with the output, outcome and impact, while the criteria of relevance ask whether all of these achievements were not only done right but were the right things to do. Finally, sustainability analyses the long-term viability of efforts.

**Figure 2 ijerph-21-01682-f002:**
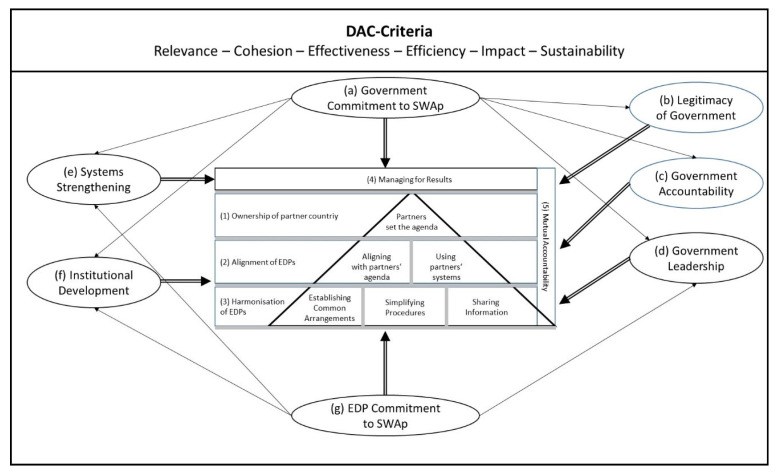
Evaluation framework. Source: author, based on [24].

The interview guideline followed the structure of the DAC criteria but left sufficient space for the interviewees to stress facts that we did not consider before, i.e., we conducted semi-structured interviews that were recorded by taking detailed notes on a laptop. For the official interviews with the Ministry of Health, all findings were summarised and presented at the end of the mission, resulting in signed minutes of meetings. The final analysis followed a simplified qualitative content analysis by Mayring with the principal categories (1–5) and (a–g) [23].

## 3. Results

In this section, we will present the findings of the quantitative and qualitative research structured by the DAC criteria as presented in Figure 2. We will add the quantitative aspects directly under the respective DAC criteria.

### 3.1. Relevance

Both programmes developed a comprehensive set of interventions for each SWAp to improve the healthcare situation and health status of the population. The SWAps entail a broad set of instruments (see also Attachment for Bangladesh). However, the analysis of the respective documents and the interviews clearly showed that a strong focus is on healthcare for the vulnerable, i.e., mother and child healthcare (MCH), health for the poor, and for people residing in peripheral regions and minorities. For both countries, it is assumed that the majority of men of the middle and upper class have comparably small problems obtaining healthcare while the situation for children, mothers, ethnic and religious minorities and the poor is much more difficult and requires strong support. Based on international and national health policies, it can be stated that these objectives and priorities are highly relevant to the health of the whole population, with a focus on the most vulnerable [26,27,28].

Furthermore, all seven SWAps included a component on health systems strengthening, which intends to improve health services by improving human resources, infrastructure, governance, etc. Based on the WHO building blocks of a healthcare system, several instruments were selected that could improve healthcare services in the long run [29].

Based on the interviews, it can be stated that the commitment of the respective governments and the EDPs was very high at least until 2020. There is a consensus on the relevance of these instruments to this day, although some interview partners have doubts about whether the SWAp is the best instrument to achieve the respective goals in the future. There is agreement that the SWAp strongly contributed to the increase in ownership by the respective government of the programme, in particular when it was combined with a basket funding mechanism. Figure 3 shows the log frame of the NHSP-II SWAp in Nepal. Based on the discussion, we can state that the main advantage of the SWAp was the increase of trust amongst development partners, as well as between EDPs and the respective government. The interviewees underlined that this trust sped up processes, reduced friction and improved the results. It also contributed to the development of ownership by the respective government. These achievements are seen as highly relevant by all interview partners. It was stated several times that trust is the foundation of alignment, ownership and good management.

### 3.2. Cohesion

The set-up of a SWAp makes it likely that all interventions and processes are consistent with national and international policies of healthcare. Based on the document analyses and the interviews, it can be stated that the SWAps reflect the objectives of the donors, e.g., WHO [30], USAID [31,32]. At the same time, we compared national strategies with the SWAp, e.g., for Bangladesh, the National Health Policy [33], the Health Care Financing Strategy (2012–2032) [34], the National Social Security Strategy [35], as well as the Bangladesh National Strategy for Maternal Health (2019–2030) [36]; and for Nepal, the Nepal Health Sector Strategy 2015/16–2020/21, the Reaching the Unreached Strategy and the Social Health Insurance Act [37]. It can be concluded that the SWAps are in line with the country’s policies, but they excluded the demand-side perspective, i.e., all seven SWAps ignored that different concepts of insurance and risk-sharing were called for by the respective governmental strategies. In the case of Nepal, a rudimentary health insurance was developed. However, the SWAps did not include this but focused on the provision of healthcare services by strengthening healthcare facilities and programmes (e.g., prevention). This might change in the future, but for the SWAps reflected on in this paper, the ignorance of the demand side called for in these strategies poses a challenge.

At the core of a SWAp is the sectoral dialogue between EDPs and the respective governments, which is based on a set of committees. They have different names in Nepal and Bangladesh, but they have similar functions. There is always a committee consisting of all EDPs working within the SWAp. In Nepal, it is called the Donor Forum, and in Bangladesh the Development Partner Consortium. The group also includes partners who do not contribute to the basket but align their activities with the SWAps. Furthermore, there were regular meetings between the EDPs and the respective MoH for exchange, planning, and reviewing. In Nepal these meetings were called a Joint Consultative Meeting and Joint Annual Review, and in Bangladesh a Local Consultative Group and Annual Programme Review.

For all SWAps reflected on in this study, the World Bank was the greatest contributor. Furthermore, it also acted as administrator of the pooled fund and shouldered the fiduciary risk for most of the phases. However, this also meant that the administrative rules of the World Bank, such as procurement policies, auditing guidelines, etc., were implemented for the SWAps and were all based on World Bank standards. Some EDPs, such as AusAid and KFW, carefully expressed that these rules did not always coincide with their own legal requirements, while government officials of the partner countries complained about the high administrative workload enforced by the World Bank.

However, there is overall agreement that the commitment of the EDPs and the respective governments to the SWAps was high, resulting in alignment and ownership.

### 3.3. Effectiveness

Figure 3 shows the logic of the DAC evaluation criteria as described above. The criteria of effectiveness ask whether the interventions produced results on the outcome level as a prerequisite of the results on the impact level. All SWAps define outcome indicators which were regularly analysed in the respective committees described above. For instance, the HPNSDP consisted of 32 operational plans (OP), which were implemented by the respective directorates of the Ministry of Health with corresponding outcome indicators (OI) [38,39]. In total, the HPNSDP had 158 indicators for 32 operational plans in the two components. In a final report on this SWAp, the World Bank concludes that 65% (102 indicators) were fully achieved and 20% were partly achieved. Merely 15% (24 indicators) were hardly or not achieved at all. It must be noted that the majority of indicators that were not achieved are in the fields of health system development, in particular personal management, development of physical infrastructure, healthcare financing, documentation and data management. Another area that was not satisfactory was the reduction of spatial disparities [38,39].

Table 2 shows some examples of outcome indicators of HPNSDP which can be used to assess the effectiveness of the respective SWAps. The contraceptive prevalence rate (CPR) objective was not achieved, while the other examples show success. When comparing the different outcomes of the SWAps, it can be seen that the number of indicators within one country increases from SWAp to SWAp. At the same time, the strategic objective (“improve the health of the population”) and the majority of indicators in Nepal and in Bangladesh are quite similar. For instance, the NHSP-II focused on “modern contraception methods”, while the HPNSDP incorporated the outcome of any contraception. Nevertheless, the utilisation of contraceptives is a major objective of all SWAps.

**Table 2 ijerph-21-01682-t002:** Examples of outcome indicators for effectiveness of HPNSDP. Source: [2,38].

Indicator	Value 2011	Objective	Latest Value
Use of contraceptives (CPR; % women in reproductive age (15–49 years) [%]	62%	72%	63% (2019)
Share of safe deliveries (births with professional support) [%]	26%	50%	59% (2019)
Vitamin A supplement (share of children from 6 to 59 months receiving vitamin A during the last six months) [%]	83%	90%	97% (2020)
Breast-feeding (share of children ≤ 6 months fully breast-fed) [%]	43%	50%	63% (2019)

Until the end of the NHSP-II and the HPNSDP, the financing of the SWAps was completely input-based, i.e., the respective government disbursed funds. Afterwards, the EDPS refunded the Ministry of Health based on the presentation of invoices submitted and the indicators fulfilled. The NHSS and the HPNSP introduced an element of results-based financing with so-called “disbursement-linked indicators” (DLI). The EDPs released funds when certain targets were met, such as a vaccination rate, successful audit, safe delivery rate, etc. (compare also Attachment for Bangladesh). As no final report on HPNSP is available yet, a concluding assessment is not feasible. However, the interview partners in both countries indicated some dissatisfaction with this approach, stating that DLI define a result irrespective of whether the achievement can be influenced by the interventions. The chain of input–output–outcome–impact is not linear or deterministic. Instead, many other variables influence the results. The contribution of the interventions of a SWAp towards defined outcomes (and finally impacts) cannot be easily assessed.

### 3.4. Efficiency

The assessment of a sector-wide activity over many years is cumbersome. However, certain proxies can be used to evaluate whether resources have been wasted. For most SWAps reflected on in this paper, the World Bank monitored and supervised the basket as a joint-donor trust fund. All payments were made based on standardised procedures, i.e., the joint financing arrangement between an EDP and the respective government stipulated that payments were made based on refunds for expenditures already made by the government on a US$ account without interest. For these services, the World Bank received a fee from EDPs between 2.5 and 3.5% of their contribution. The interview partners stressed that the transaction costs of independent programmes would have been much higher, i.e., the SWAp increased the efficiency of development assistance. None of the interview partners voiced that the administration of SWAps would be too expensive.

Another proxy is the technical efficiency of healthcare facilities. As the SWAp includes all facilities as main recipients of funds, their efficiency is of the highest relevance for the entire sector. We visited a set of representative healthcare facilities on all levels, from village clinics to tertiary hospitals in both countries, and interviewed the respective leaders. It was found that buildings and equipment are generally used according to their intended purpose, but maintenance is poor. The budget allocated to the maintenance of equipment is too low (in all healthcare facilities visited, <1% of the annual total budget), so several buildings are in poor condition and will face a premature end of useful life. However, once funds are accessible, civil engineers and craftsmen are available on the private market in both countries.

The budget for repairing the equipment seems to be higher, but two major challenges can be observed. Firstly, bio-technical engineers capable of maintaining and repairing more sophisticated medical equipment are rare and hardly available on the open market. As part of the NHSP-II in Nepal, respective staff was trained and employed by the government, but this was discontinued when the NHSP-II ended. Secondly, there is hardly any preventive maintenance. There is some effort (and funding) for repair once equipment is out of order, but the prevention of damage and breakdown is a bottleneck in both countries. The interview partners unanimously confirmed that this is not only a question of funds and personnel, but also of tradition and mentality.

Based on reports of the respective ministries and EDPs, it was expected that the healthcare facilities would be grossly understaffed. Indeed, all leaders of healthcare facilities visited confirmed that they would need more staff. However, we calculated the number of patient contacts and the occupancy per professional and found that quite a number of institutions were sufficiently staffed. Some interview partners responded to this figure by pointing out that this is a recent development, i.e., the situation has improved. There is, however, a strong disparity between urban and rural, or central and peripheral regions. Remote hospitals, for instance, have major problems in attracting doctors, while institutions in cities do not face similar challenges. One reason that was frequently given to us by the healthcare facility leadership is the fact that doctors in state healthcare facilities can only find a second job with higher income (“moonlighting”) in the city, while doctors in rural healthcare facilities can hardly survive with their government income.

We conducted a spot survey of patients in three small hospitals in Bangladesh, which found that 90% of interview partners felt that the healthcare facility had adequate staff to support them, while only 60% stated that the healthcare facility had sufficient drugs and examination facilities. Drug shortages in particular are an issue. While the availability of drugs, medical materials and vaccines has generally improved, stockouts even of essential drugs still exist, particularly in primary care facilities in both countries.

Management training and capacity are a bottleneck in many healthcare institutions, particularly in rural areas. Management is still entirely in the hands of the medical doctors. In Nepal, a new cadre of administrators is being trained, but it is still too early to assess whether this will have an impact on the efficiency of the healthcare facilities.

In both countries, we found that the mobility of the population has increased over the last 20 years. Roads are getting better, and the availability of cheap transport (bikes, motorbikes, tuktuk) enables patients to commute to hospitals in cities, bypassing one or more health centres. In principle, this is a favourable development, as the quality of services in urban hospitals is usually higher than in health centres. For instance, it allows mothers to decide to deliver in a hospital with an operating theatre rather than in small health centres without any emergency support. However, the SWAps invested a considerable amount in the development of small healthcare facilities close to the people, in particular in rural areas. The number of curative services provided by these small healthcare facilities is declining. We found that some health centres performed fewer than 50 deliveries per year, which is insufficient to ensure adequate quality. Consequently, some of the healthcare facilities which were established with funds from SWAps will have to be closed/shut down.

At the same time, the number of patients with chronic–degenerative diseases (e.g., diabetes, hypertension, et al.) is increasing. They need supervision and medication for which they do not have to travel to a larger city. The transformation of rural health centres into support centres for chronically ill patients has not started and will be the subject of future programmes. Currently, the interview partners agreed that some of the funds that were invested primarily in rural delivery facilities could have been invested more efficiently in more centralised services. However, there was also a consensus that the situation in different regions of the countries might differ; for example, public transport is less of a challenge in southern Nepal, whereas the situation in the north, with high mountains and deep valleys, needs to be assessed differently.

### 3.5. Impact

The overarching developmental objective (impact) of all SWAps considered in this paper is the improvement of the health of the entire population. Figure 4 shows some health indicators for the example of Nepal. For both countries, the trends are very positive. In particular, the mortality rates (infant, <5-year-old, neonatal and maternal mortality rate) declined tremendously from the beginning of SWAps until today. These are “hard” indicators measuring different aspects of the total mortality, with high relevance for the life and quality of life of the population. Bangladesh and Nepal have made extraordinary progress regarding these indicators, which have also been recognised, e.g.,: “Bangladesh was selected as an Exemplar due to rapid reductions in neonatal and maternal mortality rates. Bangladesh had the fastest decline in neonatal mortality of any country in the South Asia region, and the speed of its decline in maternal mortality is comparable to other neighbouring Exemplar countries, such as India and Nepal” [40].

Another highly relevant indicator is the total fertility rate. Both countries are densely populated, with Bangladesh having the highest population density worldwide (excluding city-states). However, the total fertility rate has neared the reproductive value. While the population is aging, it will still grow, but the tremendous population growth of the past with its huge challenges for the social system has come to an end.

In Bangladesh, nutrition was a major issue, so the reduction of different forms of malnutrition (stunting, wasting and underweight) were included as impact indicators. Generally, these components of the SWAps were not as successful as other interventions. The interview partners assumed that nutrition is very complex and involves many traditional beliefs and cultural habits.

For both countries, the interview partners concluded that the commitment and leadership of the government was high, resulting in ownership and alignment. At the same time, it was realised that a sector-wide approach is an instrument that is tailored to the coordination between the central government and EDPs. During the last SWAp, Nepal has gone through a federalisation and decentralisation process, resulting in much higher autonomy of regions. Consequently, the coordination and alignment between the central government, EDPs and several regional governments has become more difficult. Whether this had an impact on the respective indicators cannot be analysed. Some interview partners (from the central level) had the “feeling” that it did, but this could not be verified by the data.

**Figure 4 ijerph-21-01682-f004:**
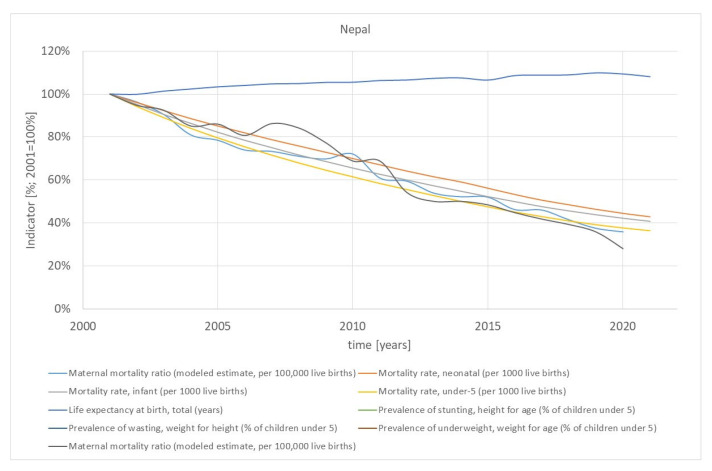
Health indicators of Nepal. Source: [2].

The greatest problems of the healthcare system of Bangladesh are still regional and social disparities. Location (urban/rural), region (divisions), education and wealth/income determine the most relevant social and health indicators. In principle, the situation of the urban population (in particular in Dhaka and Khulna) is better than in rural and remote areas, while the situation of the poor and people with lower education is challenging. This is not surprising, but the range between the quintiles is still considerably high. For instance, the share of safe deliveries of the richest 20% of the population is more than 90%, and of the poorest 20% less than 50% [41]. Similar results can be observed for Nepal, but here the impact indicators differ strongly between ethnic groups, religions and castes. For instance, the rate of teenage pregnancies of Muslims is much higher than that of Brahmin/Chhetri (15.5% vs. 5.7%). The lowest caste (Dalits) has the highest total fertility rates, and the Dalits in Terai/Madhesi have a <5 mortality rate which is 2–3 higher than the country average. In the same region, malnutrition is severe for this group, where low caste, poverty, poor education and remote location frequently coincide [42].

At the same time, the situation has improved. For instance, the range of the total fertility rate in rural and urban regions was 0.5 in 2011, and in 2022 only 0.3. Between divisions, the range was 1.2 and decreased to 0.7. The respective figures for the wealth quintiles were 1.2 and 0.8 [41,43]. Consequently, we can state that the situation has improved but is still challenging. Umesh et al. found similar results for Nepal, where the statistics remain challenging for the above-described population groups [13].

### 3.6. Sustainability

The interview partners agreed that the development of the healthcare sectors and the capacity of the respective governments to sustain the efforts of improving the health situation of the population have developed positively. The healthcare budgets in absolute and relative terms increased during the period of the SWAps, as well as the number of professionals in ministries and healthcare facilities. The phases of SWAps built on each other without interruption, and in two cases the SWAps were prolonged in order to allow a smooth flow. The harmonisation and coordination mechanisms continued and were well utilised. At the same time, the governments passed a number of relevant strategies and policies during the period of SWAps, in particular the health financing strategies.

However, the EDPs and government officials stated several times that SWAps will be phased out in the next few years, i.e., the current SWAp in Nepal and the next SWAp in Bangladesh will be the last SWAps. Cooperation and coordination are intended to continue, but the components of the existing SWAps will be taken over completely by the national governments, while EDPs will focus on new, innovative challenges, such as climate change and health. It seems that the era of SWAps will come to an end soon in these countries.

We asked the interview partners for reasons. Firstly, it was stated several times that it is a positive development that SWAps seem to be less necessary than before. The national budgets have grown significantly, so that the EDPs’ contribution is less relevant. Secondly, both countries have improved their managerial capacities so that they need less technical assistance. Thirdly, both countries have now become lower-middle-income countries. This entails that external support is not given anymore as a grant but as a loan, which is rather unattractive in the healthcare field. However, there is also some disappointment with the SWAps. In both countries, it seems that the cooperation worked best in the years from 2012 to 2018. In the first phase of the SWAps, it took some effort to make the processes work and get used to the cooperation. Afterwards, everybody seemed to be satisfied with the SWAp. One interview partner called it the “golden age of cooperation”. However, this period came to an end with major changes in personnel in the EDPs and the respective ministries.

The interview partners could not state why—in both countries—quite a few key personnel moved in 2019. For the ministries and EDPs, shifting of positions is quite common. But it seems that the cooperation and trust was discontinued in this year. There would have been a great chance to build up this trust again based on the existing formats of exchange and alignment, but then COVID-19 put an end to all efforts. Both countries suffered severely under the pandemic, with long periods of lock-down and home-office working. New staff came, but they did not meet in person for a long time. When physical meetings were feasible again, it was difficult to build up the commitment and unity that had been there before. The interview partners included officials who had been in the sector of the respective country for a long time in different positions. Several of them stated that this feeling of “my SWAp” never came back after the COVID-19 pandemic.

## 4. Discussion

### 4.1. State of the Art

Based on the document analyses and the interviews, it can be stated that the SWAps reflected on in this paper have followed the standards given by international development cooperation [4,44]. Consequently, the set-ups of the SWAps in both countries are quite similar, although components and foci differ. At the core of all SWAps in Bangladesh and Nepal is the collaboration between EDPs as well as between EDPs and the respective governments. This leads to alignment of activities and policies as well as ownership by the governments, as was described for other SWAps in a number of countries [6,9,45].

The basket funding is at the core of the SWAp. Certainly, some partners might form ‘single-donor trust funds’ with earmarked contributions to the SWAp, but the basket is the main instrument of coordination, harmonisation and alignment [6,8].

As early as 2009, the World Bank published an evaluation of health SWAps in Bangladesh, Ghana, Kirgizstan, Malawi, Nepal and Tanzania [3]. It showed that the SWAps in Nepal and Bangladesh were quite successful in the fields of harmonisation and alignment, while monitoring, evaluation and “stewardship” (i.e., efficiency) would require some more effort. This result was confirmed by our interviews almost 15 years later. The cohesion between partners has generally been seen as a major success for most of the SWAps, while efficiency remains a challenge.

The Ministry of Finance of Nepal analysed the SWAps in education and healthcare in 2018 [37]. It was reported that the International Economic Cooperation Coordination Division (IECCD) of the Ministry of Finance in Nepal speared the coordination and cooperation—a statement that was not altogether confirmed by our interview partners, who put more stress on the efforts of the Ministry of Health. However, it was important that Nepal had a “Government of Nepal’s Development Cooperation Policy”, which “sets out its preferences in terms of aid modalities and encourages Development Partners to harmonise their support in a given sector by setting up pooled funds and providing their assistance through Program-Based Approaches or Sector-wide Approaches (SWAp)”. In total, about half of official development assistance (ODA) (44–56% 2014/15–2015/16) was channelled through a SWAp, i.e., in Nepal, there is “still room for improvement as around 50% aid is still delivered outside SWAp module. Similarly, other health providers such as those outside the system are also not recorded here which means a sizable amount of investment is still done outside the SWAp framework in Nepal” [37]. Scientific evidence that is not published by the government of Nepal or an EDP is rare. Similarly to Nepal, Bangladesh has set up an “Economics Relations Division” within the Ministry of Finance to coordinate development cooperation activities. Based on a “National Policy on Development Cooperation”, this department is intended “to ensure that foreign assistance follows national development priorities as determined by national development plans and strategies and supports the country’s development efforts to bring benefits to the lives of the people”. The term “SWAp” is not mentioned in this policy, but it is obvious that the SWAp is fully in line with this target. The interview partners in Bangladesh mentioned the role of the Ministry of Finance more frequently than in Nepal, but this might also be coincidental.

Summarising, we can state that according to the literature and the perception of the interview partners, the health SWAps in both countries worked quite well and achieved their objectives. Nevertheless, there are some differences between the countries. For both countries, mother and child healthcare is pivotal, but Nepal puts even more emphasis on it than Bangladesh. Nepal had developed the “Aama Surakshya Programme” (Aama) to improve the health of pregnant women and mothers. It includes free institutional delivery, payment of transport costs and an incentive to go for four antenatal care visits. As early as 2005, the Maternity Incentive Scheme (MIS) was initiated, and in 2009, maternity fees were abolished nationwide. In 2012, MIS merged with the programme encouraging four annual ANC visits to become the Aama programme. It was an essential component of the NHSP-I, the NHSP-II and the NHSS [46,47]. In Bangladesh, MCH is highly relevant as well, but there is no specific sub-programme with such a focus within the SWAps.

At the same time, the population and landscape of Bangladesh are more homogenous than in Nepal, where health and demographic indicators differ more significantly between regions, castes and ethnic groups. About 82% of the population in Nepal are Hindu, 9% are Buddhist and 4% Muslims, while 91% of the population in Bangladesh are Muslim and 8% Hindu. Generally, the health and demographic indicators of Muslims in Nepal are significantly worse than those of Hindus or Buddhists [42]. However, the population group with the worst problems are the Dalits, as the lowest group in the Hindu caste system, with above-average reproduction rate (for Dalits and Muslims) and considerable obstacles with respect to contraceptives (below-average frequency of use amongst Newars, Terai/Madhesi Brahmin/Chhetri and Hill Brahmins; the Dalit caste shows the highest number of children per woman and the lowest incomes), vaccinations (vaccination rates and drop-out rates differ significantly between regions), obstetrics (comparable results to those with the use of contraceptives), child mortality (2–3 times higher than the national average among Terai/Madhesi Dalits) and undernutrition (especially strong among Terai/Madhesi Dalit and Terai/Madhesi) [13]. In particular, the Annual Report 2020/21 [48] depicts on numerous maps (e.g., of vaccination rates) how severe the disparity still is, which is expressed not only by geographical location with regard to the centre or periphery, but above all by belonging to a certain caste. Many live in Madhesi in the southwest of Nepal, a region marked by poor education, early pregnancies, low contraceptive rates, high mortality, etc. These differences in castes and religious groups are much lower in Bangladesh, so they did not have to be addressed by the SWAps. However, spatial disparities between urban and rural places as well as between provinces exist in both countries and were addressed by the respective instruments of their SWAps.

These findings are in line with experiences from other countries. The concept promising “better health in developing countries” [49] arose in the 1990s. It was implemented in many low- and middle-income-countries, such as the African countries of Malawi [50], Mozambique [51], Tanzania [52] and Kenya [53], as well as the Asian countries of India [54], Kyrgyzstan [55] and Cambodia [56]. The respective consequences were analysed on the micro [6] and the macro level, in particular with respect to universal health coverage [8,9]. Some countries are, however, seen by some as less appropriate environments for this mechanism, such as Mongolia [57]. It is generally agreed that the SWAp is not a “magic bullet”, but pros and cons have to be considered, and the “right time” of implementing a SWAp must be reflected on [58].

### 4.2. Advantages and Disadvantages of SWAps

Based on the analysis and interviews, we can underline a number of advantages of SWAps. Firstly, a major advantage (as agreed upon by all interview partners) is the coordination, harmonisation and alignment of partners and activities, allowing duplicate activities to be avoided and otherwise underserved areas not to be overlooked. Firstly, this results in higher efficiency, as no efforts of different partners are wasted on the same or competing interventions. Secondly, this leads to a higher ownership by the respective government. The respective ministries, predominantly the Ministry of Health and of Finance, sit in the driver’s chair and have full authority from the beginning; they have the final say and are responsible for the results. This significantly increases the chance of identification with the programme and its components, and it becomes obvious that development aid is only a short- and medium-term solution. In the long run, the government of the partner country will be fully self-sustaining and provide healthcare services for its people with its own funds and managed by its own professionals.

Thirdly, the SWAp safeguards that the priority of the country is followed, not that of the external development partners. There are certain “fashions” even in the development field, i.e., priorities of the external partners can influence the healthcare situation in the partner country. A SWAp ensures that the healthcare activities are aligned with national policies and reflect the value system of the country. Needless to say, this is not always the case, but the interview partners saw this as a major advantage of SWAps, at least increasing the likelihood of following national priorities.

Fourthly, a SWAp is a learning system. In Nepal and Bangladesh, the smooth flow of consecutive SWAps included a learning experience in the sense that components that were missing in one SWAp were frequently taken up by the following SWAp. For instance, emergency care and telemedicine were not included in the NHSP-II but became components of the NHSS. Similarly, the demand-side components (health insurance) were completely ignored during the HPNSDP but included, at least in a rudimentary manner, in the HPNSP. During the annual review meetings in both countries, achievements as well as gaps were discussed so that the new programmes could be tailored to the perceived needs of the population.

Finally, the interview partners stated that a major advantage of the SWAp was the development of trust between the partners. This trust resulted in a smooth flow of the programme, but it also strongly contributed towards the coordination between EDPs and the respective governments during crises. Nepal experienced two major external shocks during the period of the SWAps, i.e., the major earthquakes of 2015 as well as the COVID-19 crisis in 2020–2022. The interview partners indicated that the government responded in time while extensively involving the EDPs with financial and technical advice. They stated that the coordination and channelling of aid during these crises worked well because it could build on long-term trust and well-established communication platforms.

In Bangladesh, some 850,000 Rohingya came to Bangladesh in 2017 for refugees requiring extensive support and care [59]. International support was strongly backed by existing platforms and channels based on the trust between the EDPs and the government of Bangladesh. Furthermore, Bangladesh was severely hit by COVID-19 [60], and in particular the Delta-variant wave in 2021 [61]. While there was strong agreement that the trust built up during the SWAp was instrumental during the Rohingya crisis, the interview partners did not agree on whether this was also relevant during the COVID-19 pandemic. However, all partners agreed that trust was built more on the integrity and professionality of individuals than on systems, i.e., once the key persons left the country or their job, trust had to be rebuilt. This is also in line with other authors, e.g.,: “… there are many who believe that the success in developing the SWAps has had more to do with the strength of the partnerships and level of trust that have been formed than the technical soundness of the new policies and programs” [62], and the WHO concludes that “formal agreements are not an effective substitute for good working relationships, mutual trust, and strong Government ownership” [63]. However, some analyses do not even mention the word “trust”, e.g., [24].

The experiences from Nepal and Bangladesh reflected in the interviews also underlined some disadvantages of the SWAps. Firstly, “sector-wide” does not always involve the entire sector, i.e., EDPs and governments still exclude certain elements [64]. In Bangladesh, the operating budget of the healthcare facilities was included in the SWAp, but not in Nepal, i.e., the SWAps in Nepal were “development SWAps”, not “operating SWAps”. Furthermore, some EDPs still do not cooperate with the SWAp. This is frequently the case with NGOs, which do not always align their activities with the government system, but also major players such as USAID or the Global Fund, which have their own rules, not allowing a basket contribution or a global contribution to the healthcare budget. The main reasons for this policy are either that their funds are earmarked for a very specific purpose (such as from the Global Fund) or that the respective government of the EDP does not trust the national monitoring system and consequently does not allow any channelling of funds outside its own audit system (e.g., USAID). This finding was also reported by other SWAps, e.g., “notable donors including US government and the Global Fund did not participate in the SWAp, and increased vertical funding weakened the SWAp in favour of non-governmental organisations (NGOs)” [51].

Secondly, SWAps have a tendency to postpone or hinder innovations in healthcare. As the SWAp takes 2–3 years to develop and runs for at least 4 years, the technology of the SWAp is up to 7 years old. The NHSP-II, for instance, did not include any interventions against chronic–degenerative diseases, although the country entered the third phase of the epidemiological transition with a dominance of chronic–degenerative disease even before the NHSP-II started, i.e., chronic–degenerative diseases dominated cases of death due to infectious diseases in 2003. For loss of quality of life (DALYs), the respective break-even was in the year 2009 [65]. In 2019 (latest figures), 71% of death cases were due to NCDs (non-communicable diseases), 21% due to CMNNs (communicable, maternal, neonatal and nutritional diseases) and 8% due to accidents [65]. However, the NHSP-II focused only on CMNNs.

Generally, it can be stated that innovations, such as NCDs, telemedicine, insurance, emergency care, etc., take quite a long time to find their way in the SWAp. There is little evidence, and our interview partners did not contribute to this debate, but based on general innovation theory [66], it might be guessed that the disadvantage of delayed inclusion of an innovation might be compensated for at least partly by a greater adoption speed once the innovation has entered the SWAp, as it might be implemented throughout the country in the entire healthcare sector. But this would require more research.

Thirdly, from the perspective of the EDPs, a SWAp has the disadvantage that its monitoring and evaluation are crucial but difficult. An individual project or programme is usually monitored and audited by the standards of the respective donor and their government; for example, a project of the German Financing Cooperation implemented by Kreditanstalt für Wiederaufbau (KfW, Germany) follows German regulations of procurement, accounting and auditing. A SWAp consortium must agree on one specific standard. This is frequently the standard of the receiving country or the EDP with the highest budget. In the case of the SWAps of Nepal and Bangladesh, the World Bank was the strongest contributor to the basket and it set the rules and shouldered the fiduciary risk. Consequently, other EDPs had to follow World Bank regulations even if they were not identical with their own approach. Irrespective of whether the entire financial management, evaluation and auditing are done by the partner government or by one EDP, a SWAp always means that most EDPs give up some degree of control. Without trust, this is not possible.

Fourthly, providing funds via a SWAp does not guarantee that these funds can be used efficiently by the partner. In some cases, the ability to absorb and use these funds can be so poor that a major share of the SWAp funds remains unused [64]. The “absorption capacity” of the Ministry of Health was analysed by the government of Nepal. It was as low as 76% in the financial year 2010/11 [67], but it increased up to 84% in 2018/19 (latest figures). The interview partners from Nepal stated that the SWAps were generally somewhat over-budgeted, so that the lack of absorption may not necessarily be the result of weakness in implementation management, but rather of caution. Consequently, from the very beginning, it was infeasible to spend the entire fund [68].

The absorption rate of the SWAps in Bangladesh seems to be higher than in Nepal, but we only have data for HPNSDP, showing that on average the expenditure was some 93% of the original budget. However, the interview partners underlined that there was a strong delay within a financial year, i.e., delayed activities and expenditures at the beginning of the year were followed by a strong increase in procurements and payments towards the end of the year. More important is the fact that the absorption rate differs between the operational plans. While the rate was high for the control of infectious disease, physical facilities development and pre-service education (96%, 97%, 97%), it was poor for human resource management, management information systems and drug administration and management (58%, 68%, 58%). Most alarming is the fact that less than half (48%) of the funds for health economics and financing were spent as expected [39]. This reflects the impression of the interview partners that the rudimentary attempt to initiate a health insurance scheme for the poor in Tangail district (SSK, Shasthyo Shurokhsha Karmasuchi) [69] was never fully integrated into the SWAp. Similar experiences of low absorption rates within SWAps were described elsewhere [8,70], partly due to the low administrative capacity of the ministries [64].

### 4.3. “Best” Time for SWAps

Based on these pros and cons of the SWAp, one might ask whether there is a “best” time for a SWAp. During our discussions, it was stressed by a number of interview partners that there had been a “golden age” of SWAps, lasting roughly from 2012 to 2018. In both countries, it is clear that there will be no more SWAps in the long run. There seems to be general agreement that this will be the last phase of health SWAps. It is already difficult to find more pooling partners. The majority of EDPs would like to keep the coordination mechanism of the SWAp, but channel their own funds into specific programmes outside the basket. In Nepal, the new SWAp (National Health Sector Support Programme, NHSSP) could attract only FCDO (Foreign, Commonwealth and Development Office, UK) and GAVI (The Vaccine Alliance) as pooling partners; in Bangladesh, the process is still ongoing and cumbersome. The interview partners from the government stressed that they do not see a need for a SWAp anymore, as their own contribution is dominant now. The EDPs stress the fact that they would like to focus more on innovations which are not so much the focus of the partner government. For instance, climate change and health seem to have a higher priority among EDPs, while this is not on top of the priority list of the partner countries.

Figure 5 sketches the capacity of partners as well as the advantage of SWAps. In the beginning, the financial and managerial capacity of the respective government are both low. They would not be able to install and maintain a SWAp within their system. In comparison to the local contribution, the financial and technical assistance of EDPs is quite high. Consequently, a SWAp is infeasible. However, with the growing financial and managerial capacity of the government, SWAps become possible. For the time being, external partners still prefer independent programmes, because the efforts of monitoring the SWAp seem to be too high. With a growing capacity, however, SWAps become efficient and feasible and should be fostered.

Over the years, the government’s managerial capacity grows considerably, so that external technical support will generally no longer be needed. At the same time, the share of the government budget in the entire health sector increases, while the contribution of the EDPs becomes more and more irrelevant. Finally, the SWAp entails considerable coordination effort without stronger relevance for the sector, so the SWAp becomes inefficient and should be given up. EDPs can still support programmes or even projects that are generally aligned, but without the strong effort of coordinating every step. This last phase is the step towards full independence.

Bangladesh offers a good example of this development. The country started the first SWAp in 1998 with 62% government contribution. As stated before, it was a comprehensive SWAp including the operation costs of running healthcare facilities. In HNPSP (2003–2011), the government contribution increased to 67%, in HPNSDP to 76% (2011–2016) and in HPNSP to 84% (2017–2014). The successor programme, which is supposed to be the last SWAp, will have an even higher government contribution. Likewise, the government’s contribution towards the healthcare budget in Nepal increased from 58% (2010/11) to 79% (2018/19). As Nepal did not include the operational budget in the SWAp, it is appropriate to analyse this statistic for the entire healthcare budget [16,67]. However, the years 2019/20 and 2020/21 were exceptional due to a strong flow of financial assistance because of the COVID-19 crisis.

At the same time, the interview partners (with very few exceptions) agreed that the managerial capacity of the governments of Nepal and Bangladesh strongly improved. The public administration is capable of managing its own funds without (major) technical support from EDPs.

This analysis demonstrates that it was the right decision to initiate SWAps in both countries. They helped to improve the health situation and to build up the managerial capacity of the respective ministries. However, this does not mean that SWAps must be prolonged indefinitely. It is likely that SWAps will come to an end. This will not mean the end of cooperation, but external aid will be focused more on very specific issues, in particular on innovations.

### 4.4. Limitations

This analysis must be seen in the light of major limitations. Firstly, official statistics of low- and middle-income-countries such as Nepal and Bangladesh are not always reliable. International statistics, such as those from WHO and World Bank databases, rely on these national statistics. Thus, some data used in this paper could be challenged.

Secondly, it is based on documents and interviews in two countries. Thus, further research would be required to prove whether these findings are representative of other countries as well. It could be that the era of SWAps is still highly relevant for Sub-Saharan African countries while it approaches its end in South Asia. But this is beyond the scope of this paper. Thirdly, the majority of conclusions are based on interviews with officers of EDPs, ministries and healthcare facilities. We tried our best to select these interview partners thoroughly, but we cannot be absolutely sure that our interviews are not biased. In a number of cases, interview partners and healthcare facilities to be visited were recommended by government officials. It is likely that these were not the most critical stakeholders.

Finally, COVID-19 had a strong impact on the healthcare sector in both countries. The external funding strongly increased. At the same time, the prolonged lock-downs and era of home-office working paired with major changes of personnel at EDPs and even ministries resulted in a loss of social capital. Before the pandemic, partners knew each other quite well. Afterwards, trust had to be rebuilt. In the case of Bangladesh, this seemed to work rather well, while in the case of Nepal, collaboration in the year 2023 (when we visited the country and conducted the interviews) was still perceived as worse than before the pandemic by those few who had been on the job for several years. Whether this social capital can be built up again cannot be assessed today and calls for exploration in future research.

## 5. Conclusions

The SWAps in Bangladesh and Nepal addressed the right health challenges and improved the cohesion between EDPs and between EDPs and the respective governments. They were effective and efficient and had an impact on the health situation of the countries. However, their sustainability can be challenged due to factors mainly outside the SWAp, such as the COVID-19 pandemic. The majority of interview partners conclude that coordination, alignment and harmonisation improved during the era of SWAps. However, it seems that the next SWAps will be the last ones in the healthcare field. The respective governments do not need close coordination with EDPs anymore because they are almost self-governing and self-financing. One alternative would be general budget support beyond the healthcare sector, but many development partners fear the risk of limited control over the funds provided. An alternative is the shift towards some assistance in narrow and specialised programmes of innovations, including policy-based financing mechanisms to overcome structural weaknesses of the partner countries. The future role of EDPs might not be so much the financial and technical assistance for standard programmes, but the development of innovation seedlings with the potential to be included in the health sector under the control of the respective government.

## Figures and Tables

**Figure 1 ijerph-21-01682-f001:**
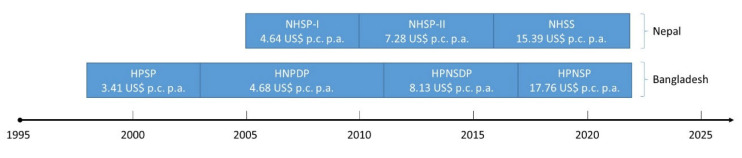
Timeline of SWAps in Nepal and Bangladesh. Source: author.

**Figure 3 ijerph-21-01682-f003:**
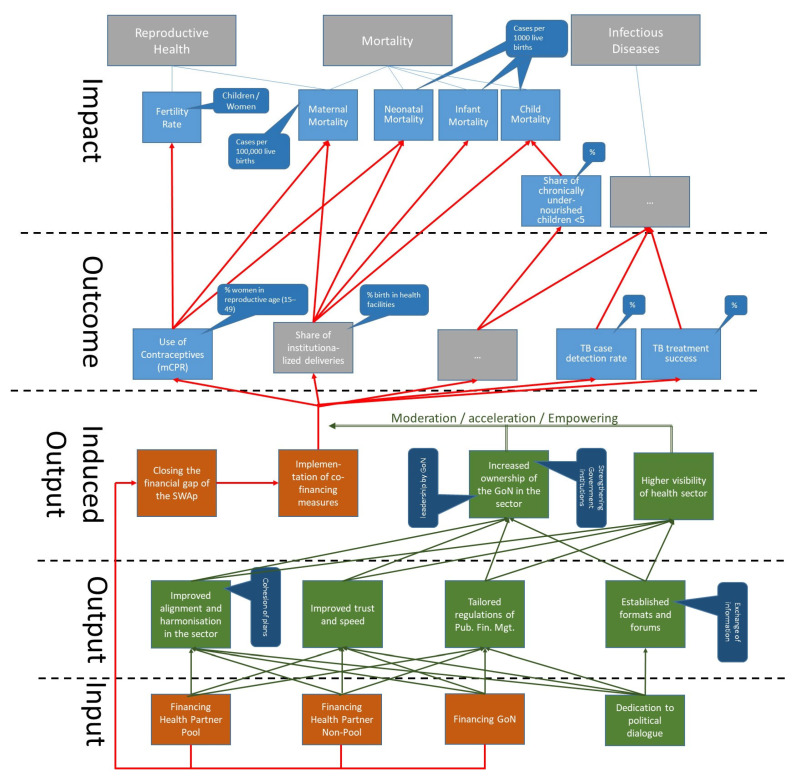
Log frame of NHSP-II. Source: author.

**Figure 5 ijerph-21-01682-f005:**
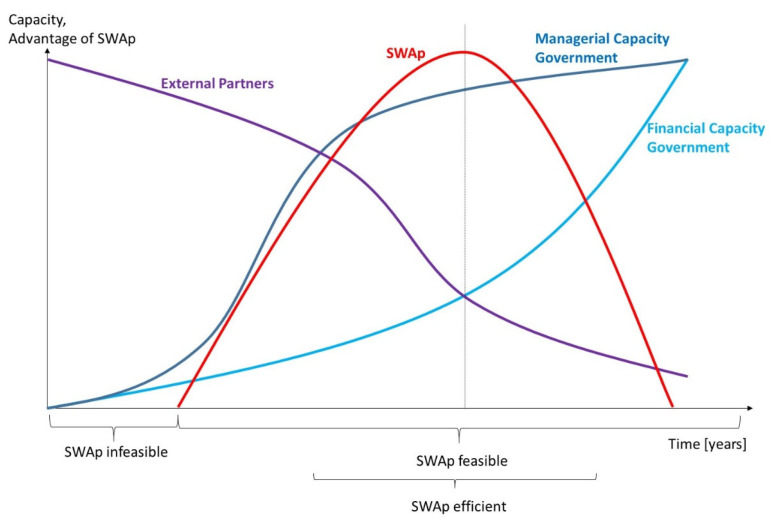
Capacities and SWAp. Source: author.

## Data Availability

All data are accessible as given under “references”.

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
