# Peer review of "Sector-Wide Approach (SWAp) in Healthcare—A Mixed-Methods Assessment of Health SWAps in Nepal and Bangladesh"

_ijerph, 2024, doi:10.3390/ijerph21121682_

Round 1
Reviewer 1 Report
Comments and Suggestions for Authors
The paper is extremely long. It should be divided into at least two papers. One way to decrease the length would be to divide the paper into two separate papers. Each of the two papers can discuss each of the countries (i.e., Nepal and Bangladesh).
Overall there are a lot of issues in terms of grammar and formatting. The paper needs to be proof-read thoroughly. The authors might benefit from working with a medical writer/ free lance writer. I have provided specific comments for several paragraphs, which were at the beginning of the paper (please see below). I hope the authors will be able to successfully use these examples to revise the draft. I am also attaching a copy of the paper and have high-lighted the sentences and paragraph that require revisions in yellow.
Several sentences in the introduction section needs improvement. For example,
Para 1, lines 9 and 10, the meaning of the sentence is not clear. Please re-phrase.
Para 2, line 1 – is a typo. Please re-phrase.
Para 3, lines 2 to 6 – appears to be a direct quote – but proper reference is not provided. This entire paragraph needs to be re-written, because meaning is not clear.
Para 6, last paragraph before methods, needs to be re-phrased. For example, what do the authors mean by next section? The last sentence needs to be removed/ deleted.
Methods:
Para 1, line 1, who did the authors survey? Last sentence, I believe ‘cast’ is mis-spelled should be “caste.”
Para 2, line 1, health is a focus of which country? Is the focus of this paragraph Nepal? I think so, but it should be clearly stated at the beginning of the paragraph.
Para 3, last line, should be mega-city
Para 4, lines 2 to 5, needs to be re-written, because meaning not clear. What do the authors mean by first half of the population? How can children have lower income? Are the authors discussing child labor?
Para 5, is this paragraph in reference to Bangladesh or Nepal? Should be clearly stated.
Para 6, line 1 – error needs to be fixed. The meaning for this paragraph is not clear.
In the results section, the authors have mentioned the word facility several times, but have not described what facility stands for.
Generally speaking the methods and the results sections should mirror each other, and the introduction and the discussion sections should also mirror each other. However, that is not reflected in this paper. Overall, the results section is a lot easier to follow, though certain phrases and statements are still challenging to understand.

Please see comments above
Reviewer 2 Report
Comments and Suggestions for Authors
Many thanks for the opportunity of reviewing your manuscript entitled ‘Sector-Wide Approach (SWAp) in Healthcare – A Qualitative Assessment of Health SWAps in Nepal and Bangladesh’ It addresses a relevant topic, which is well within the scope of this journal. The following are my comments:
Introduction
1. No clarity on objectives of the study
2. In abstract the objective is mentioned as “This paper analyzes the development and achievements of the SWAp in healthcare of Bangladesh and Nepal”, whereas in the last paragraph of Introduction mentions “This paper provides some insights into the development and relevance of SWAps in health care by the examples of Nepal and Bangladesh.” – need more clarity
3. In methodology . authors should clarify “others” who were interviewed.(12 in Nepal and 3 in Bangladesh)
Methodology
1. Title of the manuscript says qualitative assessment, whereas methodology section mentions it as ‘mixed approach” -both quantitative and qualitative - Authors need to clarify
2. Authors need to provide more details on qualitative evidence collected through key informant interviews and focus group discussions with leaders of EDPs, ministries and healthcare facilities in both countries. Who conducted these interviews in both countries? When conducted ? How was FGDs conducted? Major themes included in FGDs etc.
Results
1. Results section is mainly based on qualitative information. Authors may use quantitative findings using secondary data analysis if possible. This would provide more clarity and authenticity of the results
2. A comparative picture of the results of SWAp in Nepal and Bangladesh should be included in results /discussion section
Discussion
Authors should provide analysis of the results with other peer countries in Asia and Africa
Others
1. Avoid using “some insights” “some statistics” etc. in the last paragraph, Introduction.
2. Please rectify the issues “Error! Reference source not found” through out the document
3. Data limitations, if any may be highlighted in limitations
Round 2
Reviewer 2 Report
Comments and Suggestions for Authors
My appreciation to authors for the final manuscript. Authors have incorporated most of my recommendations.